# PDZ and LIM Domain-Encoding Genes: Their Role in Cancer Development

**DOI:** 10.3390/cancers15205042

**Published:** 2023-10-19

**Authors:** Xinyuan Jiang, Zhiyong Xu, Sujing Jiang, Huan Wang, Mingshu Xiao, Yueli Shi, Kai Wang

**Affiliations:** Department of Respiratory and Critical Care Medicine, The Fourth Affiliated Hospital, Zhejiang University School of Medicine, Yiwu 322000, China; 22218026@zju.edu.cn (X.J.); xuzhiyong@zju.edu.cn (Z.X.); 12118412@zju.edu.cn (S.J.); 21718157@zju.edu.cn (H.W.); 3160103580@zju.edu.cn (M.X.)

**Keywords:** ALP/Enigma/LIMK kinase, PDZ, LIM, tumor, scaffold protein, signaling transduction

## Abstract

**Simple Summary:**

There are three subfamilies of human PDZ-LIM family proteins with a total of 10 protein molecules, and PDZ-LIM family proteins serve as a class of scaffolding proteins that assume the function of signal transduction. In this paper, we describe the signature structural domains and major regulatory signals of PDZ-LIM family proteins and provide an overview and discussion of their functions in various tumors and major diseases, aiming to provide directions for future disease (mainly tumor) prevention and drug development.

**Abstract:**

PDZ-LIM family proteins (PDLIMs) are a kind of scaffolding proteins that contain PDZ and LIM interaction domains. As protein–protein interacting molecules, PDZ and LIM domains function as scaffolds to bind to a variety of proteins. The PDLIMs are composed of evolutionarily conserved proteins found throughout different species. They can participate in cell signal transduction by mediating the interaction of signal molecules. They are involved in many important physiological processes, such as cell differentiation, proliferation, migration, and the maintenance of cellular structural integrity. Studies have shown that dysregulation of the PDLIMs leads to tumor formation and development. In this paper, we review and integrate the current knowledge on PDLIMs. The structure and function of the PDZ and LIM structural domains and the role of the PDLIMs in tumor development are described.

## 1. Introduction

Cell signal transduction is a complex cascade process that requires precise spatiotemporal regulation to ensure the coordinated operation of signaling events. And scaffold proteins are indispensable for the coordinated and accurate transmission of signals. Although the term scaffold is widely used in academia and not strictly defined, it is currently widely believed that scaffold proteins are protein molecules that bind to two or more proteins for signal regulation [1]. Scaffold proteins mainly function in four ways: assembling signal components, regulating the localization of signaling molecules in the cell, assisting with positive and negative feedback regulation, and protecting activation signaling proteins from inactivation [2]. In these ways, scaffold proteins increase the flexibility of signaling while making it simpler and more efficient.

PDZ-LIM family proteins (PDLIMs) are kinds of scaffold proteins. Based on genetic structure and phylogenetics, they can be divided into four subtypes: ALP (PDLIM3), Elfin (PDLIM1, CLP36), Mystique (PDLIM2, SLIM), and RIL (PDLIM4) in the ALP subfamily (Figure 1); Enigma (LMP-1, PDLIM7), Enigma Homolog (ENH, PDLIM5), and ZASP (Cypher, Oracle, PDLIM6) in the Enigma subfamily (Figure 2); and LIMK1, LIMK2, and *LIM* Domain *Only* Protein *7* (LMO7, FBXO20) [3] in the LIM kinase subfamily (Figure 3). The current structural analysis shows that all other PDLIMs do not have enzymatically active structures except for the LIM kinase subfamily. All PDLIMs mainly act as scaffold proteins through the PDZ and LIM domains. Since their discovery, many PDLIM-binding proteins have been identified, including actin, kinases, and phosphatases [4,5,6,7,8]. Most of them are involved in cancer signaling. Therefore, the role of PDLIMs in cancer has received increasing academic attention. Abundant studies have demonstrated that PDLIMs are involved in the regulation of the tumor cell cycle, proliferation, metastasis, and epithelial–mesenchymal transition (EMT) by modulating signaling [9,10,11,12,13]. In this review, we focus on summarizing the role of PDLIMs in tumor biology while describing the major signaling pathways they are involved in regulating. Overall, we sought to highlight the importance of PDLIMs in tumorigenesis and development and, by reviewing the potential functions of these molecules in the regulation of signaling responses, stimulate and help future investigators develop new therapeutic strategies for tumors by considering their functions.

## 2. Structural Features of the PDLIMs

PDLIMs include at least one PDZ and one LIM domain, which contribute to the ability of PDLIMs to regulate the spatiotemporal transduction of signals through direct or indirect binding to other protein molecules and to perform a wide range of biological functions in the cell.

### 2.1. PDZ Domain

The PDZ domain is originally a 90-amino-acid tandem repeat found in the Drosophila postsynaptic density protein PSD-95, the tumor suppressor DGL, and the epithelial tight junction protein ZO-1 (hence the acronym PDZ) [14,15,16]. The classical PDZ domain is a “sandwich” structure consisting of two α-helices and six antiparallel β-helices [17]. As one of the most common protein–protein interaction domains, more than 200 PDZ domain structures have been resolved so far, including not only individual PDZ domains but also complexes of PDZ and its ligands and the dimerization of PDZ and PDZ. The N-terminal and C-terminal ends of the PDZ domain are in close proximity to each other, and the first α-helix contained therein forms a groove with the second β-fold, allowing the C-terminal peptide of the interacting protein to be embedded therein [18,19]. Based on the characteristics of the ligand C-terminal amino acid sequence, PDZ domains are traditionally classified into three categories: Class I PDZ proteins recognize the ligand C-terminal sequence X-S/T-X-Φ; Class II proteins recognize XΦ-X-Φ; and Class III proteins recognize X-D/E-X-Φ (where X indicates any amino acid and Φ indicates hydrophobic amino acid) [20,21]. However, more studies have shown that the recognition and binding of substrates by PDZ are mainly achieved through the seven amino acids at the end, and the PDZ domain can be classified into 16 different specific classes based on the characteristics of these binding motifs [22]. Moreover, both the GLGF sequence within the PDZ and the sequence within the ligand protein affect protein–ligand binding [23]. The classical example is that the PDZ domain of Syntrophin or PSD-95 proteins can interact with the β-hairpin finger structure inside nNOS proteins [24,25]. By analyzing the PDZ domain on a large scale, the researchers found that PDZ mutants can still explicitly select ligands and function after artificial mutations, which suggests that the PDZ domain has an intrinsic ability to bind substrates and is robust under mutational loading, which may be an ideal model to support the high speed of evolution of interaction networks. The authors then investigated the co-evolution of the PDZ with ligands, but due to the differences between in vitro evolution and natural selection in biology, the authors obtained an evolutionary model with a higher affinity for ligands but lower specificity. Future studies on the structural evolution of the PDZ need to be continued [26]. Tandemly repeated PDZ structural domains enable the spatially organized and coordinated binding of proteins to specific targets [27]. It is currently believed that PDZ binds mainly to membrane proteins, cytoplasmic signaling proteins, and cytoskeletal proteins. For example, the PDZ domain can bind to the palladin protein, α-actinin, and β-tropomyosin and then localize to *F-actin* to regulate the cytoskeleton and signaling [24,28]. The PDZ domain also binds to phospholipids and enhances their ability to mediate the aggregation of membrane proteins, playing an important role in maintaining cell polarization [29]. In addition, the PDZ domain not only binds to proteoglycan receptors and affects cell migration and adhesion but also binds to CNS glutamate receptors and affects synaptic plasticity, ion channel opening, and cell surface receptor expression [17,30,31]. As a central organizer of protein complexes on the plasma membrane and an important modular protein-interacting structural domain, the PDZ domain is involved in the formation of protein complexes that mediate cellular adhesion, ion transport, and signaling, thereby regulating cellular homeostasis. Several researchers have demonstrated the involvement of PDZ domains in the regulation of cell growth, differentiation, and morphogenetic movements during development using different animal models [23,32]. Continued in-depth studies of the specific binding properties of PDZ and the specific biological functions it exhibits will provide more insights into orderly intracellular signaling.

### 2.2. LIM Domain

Compared to the PDZ domain, the LIM domain is shorter, consisting of only 50–65 amino acids, and generally contains CX2CX16–23HX2CX2CX2CX16–21 CX2(CH/D) motifs [33,34]. LIM was first discovered in three transcription factors, LIN-11, ISL-1, and MEC-3, and was named from their initials. The LIM domain is rich in histidine and cysteine and can bind two zinc ions, forming a zinc finger structure [35]. Unlike GATA-like zinc finger domains, however, the LIM domain only mediates protein–protein interactions and does not recognize DNA. In contrast to PDZ domains, the recognition motif for the LIM domain has not been clearly defined to date [36]. According to the most common classification, proteins containing the LIM domain can be divided into four categories: class I LIM homomorphic domain proteins; class II LIM kinases; class III proteins with LIM domain at the C-terminus and containing other domains; and class IV LIM domain-only proteins. Similar to the PDZ domain, LIM domains act as a “scaffold”, playing a protein-binding function and taking part in a variety of intracellular signaling functions [37]. Tight turn structures containing important Tyr or Phe residues are the “sequence code” for endocytosis of receptors on the cell surface, and the LIM domain in proteins provides molecular recognition of Tyr-containing tight turn structures. As an example, Enigma binds to tyrosine tight loops in the insulin receptor via the LIM domain, thereby facilitating the activation of signaling pathways and maintaining normal physiological function [38,39]. The LIM domain, associated with cellular mechanosensing, is a major member of the integrin adhesion site, participates in the formation of cytoskeletal complexes, converts mechanical forces into biochemical signals, and plays an important role in regulating the balance between exogenous and endogenous forces [40,41,42]. In addition, the LIM structural domain can regulate gene expression by forming transcription factor complexes with other proteins or by nucleoplasmic shuttling [43]. In conclusion, the LIM domain plays indispensable roles in the cytoskeleton, cell differentiation, and organ development [44]. Several researchers have found that LIM domain mRNA is localized to the lymphatic, muscular, and circulatory systems, but whether the LIM domain plays a role in cardiac and hematopoietic morphogenesis remains to be further investigated [45]. Disruptions in signaling caused by LIM dysfunction are an important cause of disease, especially tumorigenesis. The large number of proteins contained in the LIM domain makes it difficult to understand its function in depth. However, the LIM domain is essential for the PDLIMs, and its presence is an important reason for the diversity of functions performed by this family of proteins.

### 2.3. Other Domains

In addition to the PDZ and LIM domains, some of the PDLIMs also contain other structural domains. For example, the ZM domain, first identified in ZASP proteins, mediates protein binding to α-actinin and is required for the Z-line recruitment of proteins in cultured myoblasts [46,47]. LIMK1 and 2 contain a kinase domain that regulates actin polymerization and microtubule disassembly [48]. LMO7 contains a calponin homolog (CH) domain, which is one of the most common modules of actin-binding proteins and is highly structurally conserved. The CH domain binds to actin, tubulin, and signaling proteins, regulates the actin cytoskeleton, activates signaling pathways, and mobilizes calcium ions [49,50]. The presence of these functional motifs allows the PDZ-LIM family to play a richer role in regulating signaling, which we will not repeat here.

## 3. PDLIMs and Signaling Pathways

The PDLIMs can be involved in various signaling pathways as upstream molecules or downstream targets. Among them, the most studied are integrin-related signaling, TGF-β signaling, MAPK signaling, and NF-κB signaling pathways.

### 3.1. Integrin Signaling Pathway 

Integrins are cell surface receptors composed of α and β subunits that regulate cell–cell and cell–extracellular matrix interactions [51]. At least 18α and 8β subunits are assembled into 24 different integrins [52]. Although integrins lack kinase activity, they can recruit kinases and assemble signaling complexes on the cytoplasmic face of the plasma membrane, thereby inducing signal transduction [53]. Many PDLIMs have been found to interact with integrins and affect cytoskeletal and cell adhesion functions. PDLIM1 and PDLIM5 stabilize α5β1 integrin-α-actinin-actin junctions [54]. Moreover, α2β1 integrin can activate PDLIM1, but PDLIM1 may prevent α-actinin-1 from binding to integrin subunits and foci adhesion, directing the molecule to actin filaments and stress fibers [55]. ZASP can activate α5β1 integrins in mammals and αPS2βPS integrins in Drosophila with talin, and it can also activate integrins with vimentin [42,56]. PDLIM2 deficiency upregulates the expression of β1 integrin, leading to a loss of the normal phenotype in mammary epithelial cells [57]. Upon exposure to mechanical forces, PDLIM5 and PDLIM7 are recruited to the integrin adhesion complex, which allows tendon cells and muscles to adapt to and withstand mechanical stress [58]. LMO7 is negatively regulated by the integrin-dependent signaling p130Cas, and dysregulation of LMO7 has been associated with X-linked Emory-Dreyfus muscular dystrophy [59]. Integrins also affect the expression and activation of PDLIMs. The expression of integrin α6β4 upregulates PDLIM4 and promotes cell migration capacity [60], and integrins, through activation of LIMK/Cofilin, not only control the initial formation of filopodia (FLPs) and promote the maturation of adherence sites and cell proliferation [61] but also lead to the disruption of adhesion junctions and loss of spermatogenic epithelial cells [62]. PrPc deficiency triggers integrin aggregation and overactivation of the LIMK pathway, leading to impaired neuronal synapse formation [63] (Figure 4).

### 3.2. TGF-β Signaling Pathway

TGF-β family members include three TGF-β isoforms, bone morphogenetic proteins (BMPs), and other proteins [64]. TGF-β activates the LIMK signaling axis and affects actin-myosin contractility, which in turn triggers cytoskeletal rearrangements and alters cell migration and invasion [65]. Activation of the TGF-β/LIMK pathway in retinal pigment epithelial cells leads to proliferative retinopathy in vitro [66]. The SMAD3-SMAD4 complex translocates to the nucleus and promotes the transcription of miR-181b, a molecule that promotes LIMK/Cofilin degradation via Sema3A, leading to atrial fibrosis through actin remodeling, sodium cilia formation, increased SMA stability, and the EMT process [67]. PDLIM5 promotes the migratory invasive capacity of lung cancer by inhibiting SMAD3 degradation [68]. But in pulmonary artery smooth muscle cells, PDLIM5 negatively regulates the expression of SMC markers, and overexpression of PDLIM5 inhibits TGF-β/Smad signaling, which attenuates hypoxia-induced elevation of RVSP, RV hypertrophy, and arterial wall remodeling [69]. At a certain concentration and time, TGF-β-induced expression of LMO7 inhibits TGFβ self-induction and αvβ3 integrin transcription by regulating the stability of AP-1 protein, thus limiting extracellular matrix deposition and vascular fibrosis [70]. TGF-β upregulation of LMO7 stimulates the invasive ability of mouse ascites hepatocellular carcinoma cells [71]. Due to its osteoinductive properties, PDLIM7 recruits many bone-forming factors [72,73]. The overexpression of PDLIM7 upregulates TGF-β, BMP2, BMP-4, BMP-6, BMP-7, and BMP receptors [74,75,76,77,78,79]. For example, PDLIM7 is able to activate the BMP–2/Smad1/5 signaling pathway and induces the process of pulp/osteogenic differentiation of dental pulp stem cells. PDLIM7 mediates dendritic cell proliferation through the activation of LIMK1 upon binding to BMPRII [80] (Figure 5).

### 3.3. NF-κB Signaling Pathway

The term NF-κB is commonly used to describe the p50:p65 complex. In unstimulated cells, NF-κB is mainly located in the cytoplasm due to interactions with the NF-κB inhibitor (IkB) [81]. PDLIM1 inhibits the nuclear translocation of p65 and sequesters it in the cytoplasm, thereby inhibiting inflammatory signaling [82]. Nuclear PDLIM2 controls the overactivation of inflammatory signaling by promoting p65 degradation [83]. Similarly, PDLIM7 degrades p65 alone or synergistically with PDLIM2 [84]. PDLIM2 translocates to the cytoplasm to activate NF-κB and improve cell adhesion during cell differentiation [13]. The PDLIM2/NF-κB pathway inhibits high-fat-diet-induced hepatic lipogenesis and inflammation and protects articular chondrocytes from lipopolysaccharide-induced apoptosis, degeneration, and inflammatory damage in mice [85,86]. Not only that, PDLIM2 also promotes the degradation of NF-κB and STAT3 and inhibits the development of tumor resistance [87]. The Kaposi’s sarcoma herpesvirus (KSHV) inhibits PDLIM2, activates NF-κB and STAT3, and promotes tumorigenesis and maintenance [88]. 

### 3.4. MAPK Signaling Pathway

The mitogen-activated protein kinase (MAPK) family mainly consists of extracellularly regulated kinases (ERK), MAPK14, and c-jun N-terminal kinases or stress-activated protein kinases (JNK or SAPK) [89]. LMO7-deficient mice result in altered JNK and ERK pathway activation and ultimately cardiac dysfunction [90]. The PDLIM2 inhibition of MEK and ERK phosphorylation affects cell viability [91]. PDLIM3 and PDLIM5 increase p-p38 and thus regulate the proliferation and differentiation of chicken skeletal muscle satellite cells [92,93]. PDLIM4 promotes CCR7-driven dendritic cell migration through the JNK signaling pathway [94]. PDLIM6 deficiency increases p38 MAPK phosphorylation, which induces cardiomyocyte apoptosis and ultimately dilated cardiomyopathy [95]. In canine pulmonary artery SMCs, the inhibition of ERK MAPK or p38 MAPK suppresses LIMK2 expression and affects the reorganization of the inflammation-associated actin cytoskeleton [96] (Figure 6).

## 4. PDLIMs and Tumor 

The PDLIMs serve as a platform for protein–protein interactions and are involved in the amplification and integration of signal transduction, and their indispensable role in intracellular signaling has prompted a series of studies aimed at understanding the mechanisms of the role of PDLIMs in tumorigenesis and development. These clinical and basic studies have shown that most members of PDLIMs (except for PDLIM6) are aberrantly expressed in a range of tumors, including lung cancer (LC), breast cancer (BC), and gastric cancer (GC), and are involved in the functional regulation of pathophysiology in a variety of tumors. However, the role of PDLIMs in tumor development is tissue-specific, and they play distinct roles in regulating the proliferation and migration of tumors of different tissue origins. Their specific functions need to be further investigated. 

### 4.1. ALP Subfamily 

#### 4.1.1. PDLIM1

PDLIM1 functions mainly as a key regulatory protein for cell migration and invasion in various tumors, but its role in different tumors is inconsistent [97]. In glioma, PDLIM1 is a novel scaffolding protein for the neurotrophin receptor p75 (NTR) that promotes cell metastasis by facilitating PKA phosphorylation of the p75 S303 site to induce p75 activation [98]. Studies showed that PDLIM1 was highly expressed in BC patients’ tissue and plasma samples to promote BC cell migration and invasion. And mechanistic studies showed that PDLIM1 promotes cell polarization and migration by directly interacting with ACTN1 and ACTN4 to activate CDC42 [99,100]. However, in hepatocellular carcinoma (HCC), PDLIM1 was poorly expressed, and overexpression of PDLIM1 in HCC competed for binding with the ACTN4, resulting in the dissociation of ACTN4 from F-actin. This process effectively prevented the overgrowth of F-actin and downstream Hippo signaling activation, thus inhibiting HCC metastasis [101]. Similarly, PDLIM1 was lowly expressed in GC and promoted GC progression and cisplatin resistance [102]. PDLIM1 deletion or knockdown in highly metastatic colorectal cancer cells (CRCs) reduces the stability of the E-cadherin/β-catenin adhesion complex and inhibits the transcriptional activity of β-catenin on EMT-related genes [103]. Moreover, silencing of PDLIM1 in choriocarcinoma (CC) cells leads to cell expansion and loss of stress fibers and local adherent patches, a process that can be rescued by the overexpression of full-length PDLIM1 but not by mutants with deletion of the PDZ or LIM domain, confirming that PDLIM1 plays an important function in the pathogenesis of CC through its protein-binding domain [104]. In addition, PDLIM1 also acts as a tumor antigen to induce antibody responses in pancreatic adenocarcinoma (PAAD), but whether this immune response is tumor-specific needs further investigation [105].

#### 4.1.2. PDLIM2

PDLIM2, located on chromosome 8p21, is frequently disrupted in various cancers [11,106]. In BC, conflicting findings regarding PDLIM2 expression and function have been reported. Ding et al. found that highly expressed miR-222 targets PDLIM2 and inhibits its expression, leading to malignant progression and lymphatic metastasis [107]. Moreover, the study also found hypermethylation in the PDLIM2 promoter region, and methylation inhibitors or vitamin D treatment resulted in demethylation of the PDLIM2 promoter, leading to upregulated PDLIM2 expression and BC progression arrest [108]. Further studies showed that PDLIM2 expression was associated with the EMT process. PDLIM2 is lowly expressed in non-EMT BC cells but highly expressed in infiltrating cells that do undergo EMT. Another study revealed that PDLIM2 promoted cell-directed migration, cytoskeletal polarization, and the EMT process by activating COP9 signaling vesicle activity and NF-κB and STAT3 signaling [109,110]. In addition, it was shown that high PDLIM2 expression in BC was associated with higher M2 macrophage infiltration. 

In LC, PDLIM2 acts as a tumor suppressor gene. PDLIM2 represses multidrug resistance genes and cancer-associated genes, rendering cancer cells susceptible to immune attack and treatment [87]. Global or lung epithelial-specific deletion of PDLIM2 promotes LC development, chemoresistance, and resistance to anti-PD-1 therapies [87]. Moreover, the overexpression of PDLIM2 hinders the proliferation, colony formation, and invasive capacity of NCLCL cells by inhibiting NF-κB signaling [111].

The function of PDLIM2 has also been reported in other cancer types. In esophageal squamous cell carcinoma, PDLIM2 expression was significantly downregulated, and exon 7/8/9/10 of PDLIM2 was a novel valuable predictive characterization for esophageal cancer (EC) patients [112]. In HCC, PDLIM2 is also downregulated and associated with poorer prognosis. PDLIM2 exerts tumor-suppressive effects by inhibiting HCC cell proliferation, migration, invasion, EMT, and colony formation through the inhibition of β-catenin activity [113]. In addition, macrophage-derived exosomes from Laryngeal squamous cell carcinoma (LSCC) inhibited PDLIM2 expression by delivering miR-222-3p to LSCC cells, leading to elevated PFKL expression and enhanced glycolysis and thereby accelerating cell proliferation [114]. In ovarian cancer (OV), PDLIM2 is epigenetically suppressed, and PDLIM2 inhibition promotes OV growth in vivo and in vitro via NOS2-derived nitric oxide signaling, leading to M2-type macrophage recruitment [115]. In addition, some clinical analyses showed that PDLIM2 expression was also significantly downregulated in patients with metastatic CRC [116]. However, in prostate cancer (PC) and GC, PDLIM2 plays a pro-carcinogenic role, and specific deletion of PDLIM2 significantly suppressed cell proliferation and migration [91,117].

#### 4.1.3. PDLIM3 

PDLIM3 has been relatively poorly studied in tumors. In medulloblastoma, studies have shown that the abnormal expression of PDLIM3 and four other genes leads to the abnormal activation of Hedgehog (Hh) signaling, which is an important diagnostic marker for medulloblastoma patients [118]. A weighted gene co-expression network analysis of 401 patients with invasive uroepithelial carcinoma of the bladder revealed that high PDLIM3 expression was associated with poor patient prognosis [119]. However, Lu et al. found that PDLIM3 expression was significantly decreased in patients with bladder cancer (BCA) through a tumor cell differential expression gene screen, and the specific mechanism of action deserves further exploration [120]. PDLIM3 expression was also decreased in papillary thyroid carcinoma (PTC) samples from the Chernobyl region, but the correlation between PDLIM3 expression and clinical stage, metastasis, and prognosis of thyroid carcinoma (THCA) patients and its role in the pathogenesis of THCA is unclear [121]. 

#### 4.1.4. PDLIM4

PDLIM4 is considered a tumor suppressor gene whose expression is downregulated in a variety of cancers. For example, it was shown that PDLIM4 expression is downregulated in OV; that its expression is negatively correlated with clinical stage, lymphatic metastasis, and patient survival; and that the overexpression of PDLIM4 inhibits OV cell proliferation, migration, invasion, and xenograft tumor growth by suppressing STAT3 signaling activation [122]. Hypermethylation of the PDLIM4 gene can be used as a diagnostic marker for PC. Numerous studies have shown that hypermethylation of the PDLIM4 gene leads to the development and malignant progression of PC, and functional experiments have confirmed that PDLIM4 can inhibit tumor growth by suppressing PC cell proliferation [123,124,125,126]. In addition, the PDLIM4 gene was also found to undergo hypermethylation in THCA and kidney cancer, but whether the change has a biological effect is still unknown [127,128]. For now, the role of PDLIM4 in BC is controversial. Feng et al. found that PDLIM4 was highly methylated in tumor tissue by analyzing 38 pairs of patients with primary BC lymphatic metastases [129], and other studies showed that PDLIM4 expression was negatively correlated with tumor size, differentiation status, and SPF (S-phase fraction) value [130]. However, it has also been shown that upregulation of PDLIM4 expression promotes BC cell migration [131]. Proper assessment of its role in the malignant progression of BC facilitates the implementation of individualized tumor treatment. These findings shake the perception of PDLIM4 as an oncogene, and the complexity of life signal transduction makes it possible for the same gene to perform completely opposite functions in different settings (Table 1).

### 4.2. Enigma Subfamily

#### 4.2.1. PDLIM5

High expression of PDLIM5 in PC is significantly correlated with poor survival. And the rs17021918 mutation of PDLIM5 is negatively correlated with the progression of PC patients. Liu et al. found that PDLIM5 can inhibit AMPK ubiquitinated degradation by binding to it, thus promoting the malignant progression of PC [132,133]. Meanwhile, Yan et al. found that PDLIM5 is a novel substrate for AMPK, which can directly phosphorylate PDLIM5 Ser177, and mutations at this site significantly inhibit cell migration and attenuate lamellipodia formation [134]. Furthermore, PDLIM5 promotes THCA malignant progression by activating the Ras/ERK signaling pathway [135]. In LC, high expression of PDLIM5 accelerates metastasis by competitively binding Smad3 with the E3 ubiquitin ligase STUB1 and promoting TGFβ signaling activation and EMT [68]. Similarly, others have probed and found a trend toward better survival in NSCLC patients with low PDLIM5 expression [136]. High expression of PDLIM5 also promotes gefitinib resistance by inhibiting autophagy in PC9GR cells [137]. Knockdown of PDLIM5 through a novel mesoporous silica-loaded PDLIM5 siRNA (small interfering RNA) nanoplatform can reverse the gefitinib-resistant phenotype in PC9 [138]. 

#### 4.2.2. PDLIM6

PDLIM6 is a myosin Z family protein expressed mainly in the transverse muscle, skeletal muscle, and heart. Its deletion or mutation causes mainly cardiac and skeletal muscle disorders. To date, there is no direct evidence of a role for PDLIM6 in tumor development [139,140,141]. 

#### 4.2.3. PDLIM7

PDLIM7 plays a significant role in cancer development and progression. In acute myeloid leukemia (AML), high expression of PDLIM7 is an independent risk factor for poor event-free survival (EFS) and overall survival [142]. It is also associated with negative patient survival outcomes in BC, where it prevents the ubiquitination degradation of RETMEN2A mediated by cbl-c [143]. Furthermore, in liver metastasis BC cells, both PDLIM2 and PDLIM7 can bind to Claudin-2, a molecule known to promote BC liver metastasis, suggesting their involvement in BC metastasis [144]. 

In HCC and CRC, PDLIM7 enhances the growth of cancer cells. Mitogenic stimulating factors like serum, FGF, and HGF increase PDLIM7 transcription by inducing the serum response factor (SRF). PDLIM7 then inhibits the auto-ubiquitination of P53 by binding to MDM2, leading to P53 degradation and promoting cancer cell growth [145]. 

In THCA, PDLIM7 is aberrantly highly expressed in all subtypes and positively correlates with lymphatic metastasis and tumor malignancy. Subcellular localization analysis reveals differences in PDLIM7 distribution, with cytoplasmic localization in classical papillary thyroid carcinoma (PTC) and mixed cytoplasmic and nuclear staining in follicular papillary thyroid carcinoma (FVPTC), poorly differentiated thyroid carcinoma (PDTC), and undifferentiated thyroid carcinoma (ATC). This suggests that PDLIM7 localization may be associated with the malignant progression of THCA [146]. This suggests that PDLIM7 localization may be associated with the malignant progression of THCA. PDLIM7 also plays a role in regulating THCA progression by selectively binding to the THCA-specific RET/PTC2 mutant short heterodimer protein [147]. Knockdown of PDLIM7 in THCA cells leads to the downregulation of AKT and Survivin protein expression, inhibiting cell proliferation and promoting apoptosis [148]. 

Moreover, recent studies have identified PDLIM7 in cancer-associated fibroblasts binding to calponin 1 protein and inhibiting its degradation by E3 ubiquitin ligase NEDD4-1. This activation of ROCK1/MLC signaling leads to increased matrix stiffness and promotes the activation of YAP signaling in GC cells, making them resistant to 5-fluorouracil treatment [149]. However, a few studies have also shown that PDLIM7 exerts oncogenic functions, such as the downregulation of PDLIM7 expression in osteosarcoma (OS) tissues and the inhibition of cell proliferation and migration by PDLIM7 overexpression in OS cells [150]. 

Overall, PDLIM7 plays complex roles in various cancers, acting as both a promoter and inhibitor of tumor progression depending on the specific cancer type Table 2.

### 4.3. LMO7

LMO7 was reported to be a marker for BC diagnosis. LMO7 is highly expressed in invasive BC tissues and synergizes with Rho GTPase to activate myocardin-related transcription factors (MRTFs), which then upregulate SRF transcriptional activity to promote BC cell migration [151]. Analysis of the clinical correlation between LMO7 expression and lung adenocarcinoma (LUAD) patients showed that LMO7 expression was negatively associated with lymphatic metastasis and poor prognosis [152]. Animal experiments revealed that lacking LMO7 increased susceptibility to spontaneous LC in mice. In vitro experiments showed that LMO7 knockout resulting in numerical chromosome abnormalities was the major cause of LC induction [153]. Meanwhile, it has been shown that miR-96 promotes proliferation, migration, and drug resistance of LC cells through the downregulation of LMO7. These findings suggest that LMO7 acts as a tumor suppressor in LC pathogenesis [154]. LMO7 has been reported to be involved in tumor progression by fusing with other genes. Two case reports showed that LMO7 and ALK1 can form fusion mutations, which affect LC progression and targeted drug efficacy [155,156]. In addition. LMO7-BRAF rearrangement has been found in THCA, and the fusion protein formed stimulates ERK1/2 phosphorylation and promotes cell growth in a similar pattern to BRAF [90]. The overexpression of LMO7 in CRC cells and cervical cancer (CCA) cells interfered with spindle check site (SAC) action, prolonged mitosis, and inhibited cell proliferation [157]. In PAAD, LMO7 deficiency inhibited tumor cell proliferation and metastasis and the progression of subcutaneous hormonal tumors [158]. LMO7 expression was positively correlated with the metastatic ability of HCC, but it is puzzling that in vitro overexpression of LMO7 did not promote the migration of HCC cells [71]. LMO7 is also involved in tumor microenvironment regulation. DAPK3 induces K63-linked polyubiquitination of STING by phosphorylating LMO7, which in turn drives tumor-intrinsic innate immunity and tumor immune surveillance [159]. 

### 4.4. LIM Kinase

Of all PDLIMs, LIMK1/2 has been most extensively studied as a kinase molecule, and more studies continue to attempt to gain insight into the role of LIMK1/2 in tumor progression. The primary function of LIMK1/2 is to act as a signaling node that controls the dynamics of the actin cytoskeleton and regulates the phosphorylation of cofilin [160,161,162]. LIMK1/2 shares nearly 70% similarity in the kinase domain and binds to macromolecular ligands mainly through the LIM domain, which include Rho GTase, Rac1, Cdc42, RhoA, and their downstream effect molecules p21 activated protein kinase (PAK) 1–4 and ROCK1/2. ROCK1/2 and PAK activate LIMK through phosphorylation. Activated LIMK specifically binds to the downstream molecule cofilin and phosphorylates the Ser3 site of cofilin, leading to its inactivation. The occurrence of this event subsequently regulates cell morphology, motility, adhesion, and migration by modulating cellular actin backbone polymerization. As with all other kinase molecules, the development of inhibitors against LIMK1/2 is currently an ideal strategy for antitumor drug development [163,164]. The following sections compile the current understanding of the involvement of LIMK1/2 in tumor progression.

#### 4.4.1. LIMK1

##### LIMK1 in GC

As shown, LIMK1 is highly expressed in GC patients and positively correlates with the degree of GC differentiation clinical stage, lymph node metastasis, and poor prognosis [165]. Treatment with PAK4 inhibitors (GL-1196, LC-0882, and LCH-7749944), diallyl disulfide (garlic extract, DADS), bitter ginseng, quercetin, and knockdown of RhoGDI2 or DGCR6L have been shown to inhibit GC malignant progression by inhibiting LIMK1/Cofilin signaling [166,167,168,169,170,171,172,173]. In addition, the clinical agent Dabrafenib has also been shown to inhibit GC metastasis by targeting LIMK [174].

##### LIMK1 in CRC

LIMK1 was highly expressed in CRC tissues and positively correlated with metastasis, overall survival, and the pathological grade of patients [175,176]. LIMK1 was found to promote CRC proliferation and metastasis by binding to STK25, MYH9, and ACTIN4 [177,178]. As in GC, DADS treatment also inhibited the malignant progression of CRC by suppressing LIMK1/Cofilin signaling [179]. In addition, miR-27b-3p and miR-145 have been reported to exert antitumor effects in CRC by inhibiting LIMK1/Cofilin signaling through targeting LIMK1 and PAK4, respectively [180,181]. However, the role of LIMK1/Cofilin signaling in CRC is controversial, as recent studies have shown that IRX5 and SSH3 promote CRC progression by inhibiting this signaling [182,183].

##### LIMK1 in BC

Several studies have confirmed that LIMK1 is significantly upregulated in BC patients. The specific Rock and Rho inhibitors Y-27632 and C3 inhibit LIMK1 activation to suppress BC cell migration and invasion [184,185]. LIMK1 promotes tumor growth and tumor angiogenesis by increasing uPA expression [186]. In addition, LIMK1 alters the distribution of cortical actin in MT1-MMP positive inclusions and then promotes BC cell migration [187]. Molecular biology studies showed that EBP50, SEMA3B, MEX3A, and SphK2/S1P affect BC cell migration by regulating LIMK1/cofilin signaling [188,189,190,191]. Pharmacological studies revealed that herbal monomer serralactone A inhibits BC cell migration by inhibiting LIMK1 activity [192]. Similarly, miR-200b-3p, miR-429-5p, miR-128-3p, miR-519d-3p, and miR-143-3p inhibit BC cell proliferation and migration by downregulating LIMK1 [193,194,195,196].

##### LIMK1 in PC

The current results show that ectopic expression of LIMK1 gives PC cells an aggressive phenotype [197]. MiR-143, miR-23a, and nuclear clusterin exert cancer-suppressive effects by blocking the activation of LIMK1/Cofilin [198,199,200]. In addition, CXCL12/CXCR4 antagonizes doxorubicin-induced cell cycle arrest by activating LIMK1/Cofilin signaling [201]. In androgen-dependent PC patients, the LIMK1 inhibitor inhibits PC cell proliferation and migration by altering microtubule-dynamics-impeding DHT-induced androgen receptor nuclear translocation, protein stability, and transcriptional activity [202].

##### LIMK1 in LC

LIMK1 expression was significantly upregulated in NSCLC patients. Bioinformatic analysis revealed that LIMK1 could be used as a biomarker for poor prognosis in LUAD and a potential target for immunotherapy [203,204]. Consistent with other tumors, in NSCLC, PAK4 can also regulate cell migration and invasion by regulating the phosphorylation of LIMK1 [205]. The herbal monomers Leuconostoc and Mucuna pruriens inhibit LIMK1/Cofilin signaling and thus LC cell migration [206,207]. miR-27b inhibits the proliferation and migration of NSCLC cells by suppressing LIMK1 expression [208]. In addition, knockdown of LIMK1 enhanced the sensitivity of LC cells to cisplatin [209].

##### LIMK1 in OS

Immunohistochemical analysis of cancer and paracancer tissues revealed that LIMK1 was highly expressed in OS tissues [210]. Insulin stimulation activates the LIMK1/Cofilin pathway and thus promotes OS cell proliferation [211]. 6-Hydroxythioglycine inhibits the migration of OS cells by reducing LIMK1 expression [212]. In addition, it was shown that LIMK1 expression was elevated in multidrug-resistant OS cells and that knockdown of LIMK1 enhances the sensitivity of multidrug-resistant cells to drugs [213].

##### LIMK1 in Cervical Cancer (CC)

The expression level of LIMK1 in CC was significantly higher than that in the control and heterogeneous hyperplasia groups. Further analysis revealed that LIMK1 expression was positively correlated with metastasis and negatively correlated with patients’ overall survival [214]. It was reported that miR-125a-5p enhanced the efficacy of cisplatin on CC cells by targeting LIMK1. In addition, FOXD3-AS1 promoted CC progression by competitively sponging miR-128p to upregulate LIMK1 expression [215,216]. 

##### LIMK1 in HCC

The role of LIMK1 in HCC has been discovered recently, and studies have shown that exo-miR-374c-5p inhibits EMT by targeting the LIMK1-Wnt/β-catenin signaling to suppress HCC metastasis [217], while lncRNA H19 upregulates LIMK1 by sponging miR-520a-3p to promote HCC progression [218]. Furthermore, studies on HCC revealed that EGF drives the nuclear translocation of LIMK1 by activating the interaction between p-ERK and LIMK1 and that nuclear LIMK1 directly binds to the promoter region of c-myc to stimulate c-myc transcription, thereby promoting HCC progression [219].

##### LIMK1 in Other Tumors

LIMK1 also plays an important role in other tumors. For example, overexpression of the integrin αVβ3 inhibits RhoA activation and thus LIMK1-mediated cofilin phosphorylation, which promotes melanoma cell migration and invasion [220]. It was found that miR-106a inhibits cell proliferation, migration, and EMT by suppressing LIMK1 expression in oral cancer cell lines [221]. In addition, miR-20a and miR-373 inhibited the progression of THCA/cutaneous squamous cell carcinoma and glioblastoma, respectively, by targeting LIMK1 [222,223,224]. LINC00941 acts as a sponge for miR-335-5p, upregulates ROCK1 to activate the LIMK1/Cofilin-1 pathway, and subsequently facilitates PAAD growth and metastasis [225]. Immunohistochemical staining showed that LIMK1 was highly expressed in OV tissues and correlated with the clinical stage of OV patients. In OV cells, ET-1/ETAR promotes LIMK1/Cofilin activation through activation of Rock signaling, thereby promoting cell migration [226]. In addition, miR-138 inhibits OV cell migration by targeting LIMK1, but LIMK1 is not involved in miR-138-regulated cell proliferation [227].

#### 4.4.2. LIMK2

##### LIMK2 in CRC

Studies have shown that LIMK2 expression is downregulated in CRC patients and continues to be downregulated as the tumor deteriorates [228]. Filipe et al. further confirmed that LIMK2 is poorly expressed in the intestines of cancer-prone mice, as well as in human CRC cell lines and tumors. The reduced expression of LIMK2 is associated with shortened OS in patients. Further results revealed that LIMK2 inhibits the proliferation of stem cells, and LIMK2 deficiency promotes the growth of CRC in mice [229]. In addition, the overexpression of LIMK2 in CRC downregulates β-catenin and inhibits WNT signaling, thereby inhibiting cell proliferation and migration. However, LIMK2 has also been reported to be highly expressed in CRC tissues and positively correlated with clinical stage and lymph node metastasis [230]. Upregulated LIMK1/2 contributes to malignant progression and chemoresistance in CRC. In addition, miR-939-5p was shown to inhibit CRC metastasis by targeting LIMK2 [231].

##### LIMK2 in PC

LIMK2 is a specific target for the treatment of castration-resistant PC (CRPC), which is upregulated in androgen deprivation therapy. Inducible knockdown of LIMK2 completely reversed CRPC tumorigenesis in castrated mice. Mechanistic studies revealed that TWIST1 is a direct substrate of LIMK2. Under hypoxia, LIMK2 induces TWIST1 transcriptional activation and stabilizes TWIST1 by direct phosphorylation, thereby mediating CRPC development [232]. Recent studies have shown that SPOP, PTEN, and NKX3.1 are also substrates of LIMK1 and that LIMK1 affects PC progression and resistance by regulating the phosphorylation and degradation of these substrates [233,234,235].

##### LIMK2 in BC

Aurora A is a serine/threonine kinase that is overexpressed in most tumors. Emmanuel et al. identified LIMK2 as a substrate for Aurora A. Aurora A regulates LIMK2 enzyme activity, localization, and expression by promoting phosphorylation at S283, T494, and T505 sites of LIMK2, thereby inducing the occurrence of BC. In turn, LIMK2 promotes Aurora A expression through positive feedback regulation [236]. It has also been shown that LIMK2 promotes the metastatic progression of TNBC through the activation of SRPK1 [237]. In addition, consistent with LIMK1, SEMA3B also inhibits BC cell migration by suppressing LIMK2 activity [191].

##### LIMK2 in LC

Xu et al. found that MED12 knockdown activates LIMK2, causing abnormal remodeling of the actin cytoskeleton, disrupting the shedding of intercellular bridges, and leading to cytokinesis failure, thereby inhibiting NSCLC cell proliferation [238]. In contrast, lncRNA TUG1 upregulates LIMK2b (a splice variant of LIMK2), thereby promoting NSCLC proliferation and drug resistance [233]. Other studies on non-coding RNAs have also shown that PPVT1 and DHRS4-AS1 promote the expression of LIMK2 through miR-423-5p, thereby affecting the progression of lung squamous cell carcinoma (LUSC) [234].

##### LIMK2 in OS

When OS cells are stimulated by EGF, EGF activates RhoA and then phosphorylates LIMK2/Cofilin to promote cell migration. Knocking down LIMK2 in OS inhibits the formation of actin stress fibers and cell migration while also inhibiting the occurrence of EMT. In addition, BMPR2 and PD-L2 also regulate OS metastasis by regulating the RhoA-Rock-LIMK2 signal [235,239]. 

##### LIMK2 in Neuroblastoma

LIMK2 is highly expressed in vincristine- and colchicine-resistant neuroblastoma cell lines. And the inhibition of LIMK2 expression increased the sensitivity of neuroblastoma to vincristine and colchicine in [240]. Meanwhile, some findings suggest that LIMK2 expression is elevated in neuroblastoma cells resistant to microtubule-targeted drugs. Further, mechanistic findings suggest that LIMK2 is involved in the regulation of cellular drug resistance through the regulation of microtubule acetylation and microtubule protein polymerization protein 1 (TTP1) expression [241].

##### LIMK2 in Other Tumors

A zebrafish xenograft model showed that LIMK2 synergistically regulates PAAD development with LIMK1 [242]. In addition, the LIMK2 inhibitor T56-LIMKi inhibited the growth of subcutaneous tumor models of PAAD in mice by specifically inhibiting LIMK2 without cross-reacting with LIMK1 [243]. Wang et al. demonstrated for the first time that LIMK2 was highly expressed in BCA patients. Multifactorial logistic regression analysis showed that the SNP mutation rs2073859 (G-A mutation) in the UTR region of LIMK2 was significantly more prevalent in BCA patients than in controls, and the mutation was positively correlated with metastasis and the clinical stage of patients. Functional assays showed that LIMK2 overexpression promoted BCA cell proliferation, migration, and invasion [244]. Research has found that LIMK2b is downregulated in esophageal and THCA tissues. Overexpression of LIMK2b promotes Cofilin phosphorylation, leading to the arrest of these two types of tumor cells in the G2/M phase and a decrease in cell migration ability [245] (Table 3).

## 5. Conclusions and Future Directions

Cancer occurs when mutations or epigenetic alterations in genes that regulate signal transduction lead to the dysregulation of intracellular signaling homeostasis [249]. Many reports have highlighted the important role of scaffolding proteins as a “silver bullet” for signal transduction in maintaining these sophisticated signals. PDLIMs are several different groups of scaffolding proteins that play essential roles as mediators in various physiological processes such as cell proliferation, stemness, apoptosis, differentiation, and migration [97,250]. In this article, we highlight their role in tumor development. The level of PDLIM expression or protein dysfunction leads to tumor susceptibility and tumor progression, and the detection of PDLIM expression can effectively improve the identification of tumors. Although a large number of assays have clarified that the expression of PDLIMs is dysregulated in tumors, there is still a paucity of studies on the mechanism that triggers their dysregulation, with there being basically only miRNA-related studies in the literature, and continuing to search for upstream regulatory molecules has certain significance for inhibiting the signal at the source [251]. Considering the central role of PDLIMs in a wide range of tumors, targeting this class of molecules to synthesize corresponding inhibitors provides a new entry point for molecularly targeted therapies. To date, inhibitors targeting LIMK1 and LIMK2 have shown potent tumor-suppressive effects. Moreover, although there are no known active domains for the other eight proteins except LIMK1 and LIMK2, the emergence of the targeted protein degradation technology “PROTAC” offers the possibility of drug discovery for this class of non-druggable protein targets and is a worthy direction for future efforts [252]. 

However, as mentioned above, most PDLIMs have two-sided effects on tumor regulation, which may be related to the different protein signaling pathways interacting with each other in different environments; therefore, further exploration of the mechanism of action of PDLIMs in the process of tumorigenesis is of positive significance for the rational use and control of these molecules in tumor prevention and treatment. Meanwhile, PDLIMs of the same protein subfamily have great structural similarity, so, do they have synergistic effects among themselves in regulating tumor progression? At present, there are relatively blank studies on this point. A systematic and in-depth study of the relationship between PDLIMs and tumors will help people understand the status and role of PDLIMs in tumorigenesis, which is important for tumor diagnosis and treatment (Table 4).

## Figures and Tables

**Figure 1 cancers-15-05042-f001:**
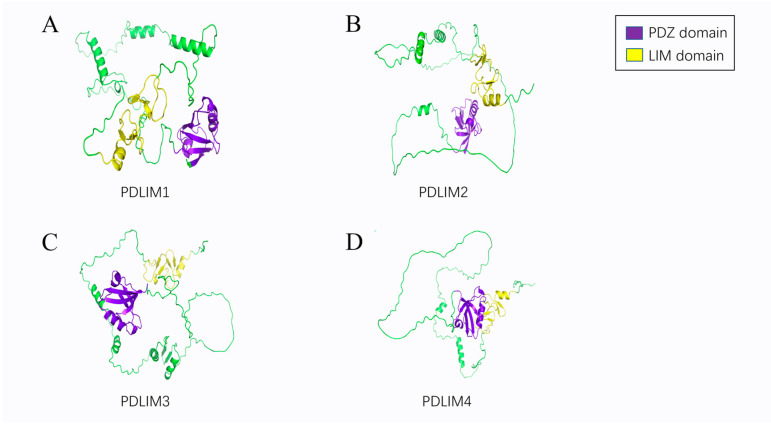
The 3D structure of ALP subfamily. (**A**) The 3D structure of PDLIM1. (**B**) The 3D structure of PDLIM2. (**C**) The 3D structure of PDLIM3. (**D**) The 3D structure of PDLIM4.

**Figure 2 cancers-15-05042-f002:**
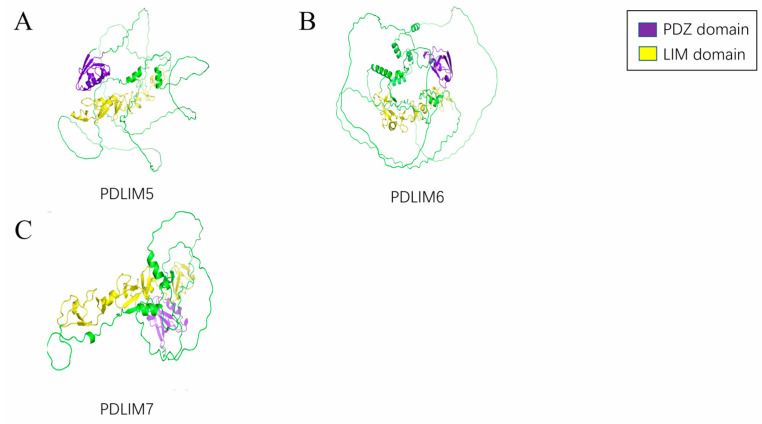
The 3D structure of Enigma subfamily. (**A**) The 3D structure of PDLIM5. (**B**) The 3D structure of PDLIM6. (**C**) The 3D structure of PDLIM7.

**Figure 3 cancers-15-05042-f003:**
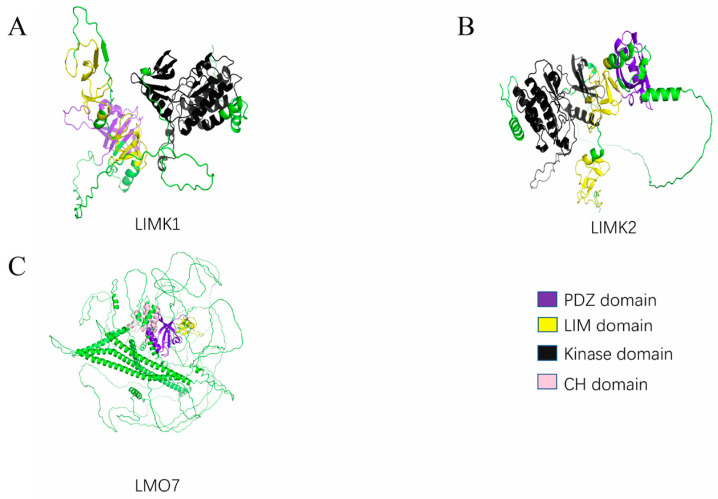
The 3D structure of LIM Kinase and LMO7. (**A**) The 3D structure of LIMK1. (**B**) The 3D structure of LIMK2. (**C**) The 3D structure of LMO7.

**Figure 4 cancers-15-05042-f004:**
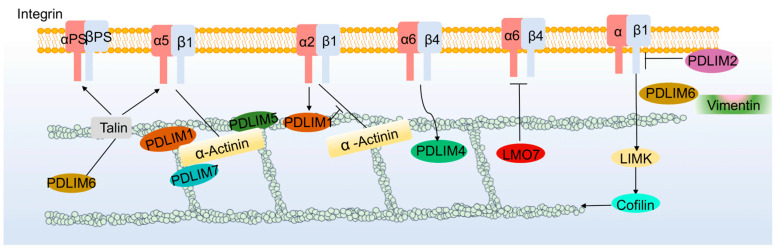
PDLIMs and integrin signaling pathway.

**Figure 5 cancers-15-05042-f005:**
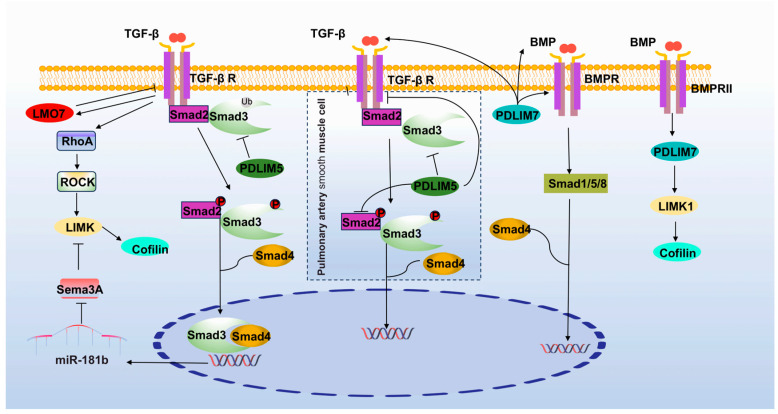
PDLIMs and TGF-β signaling pathway.

**Figure 6 cancers-15-05042-f006:**
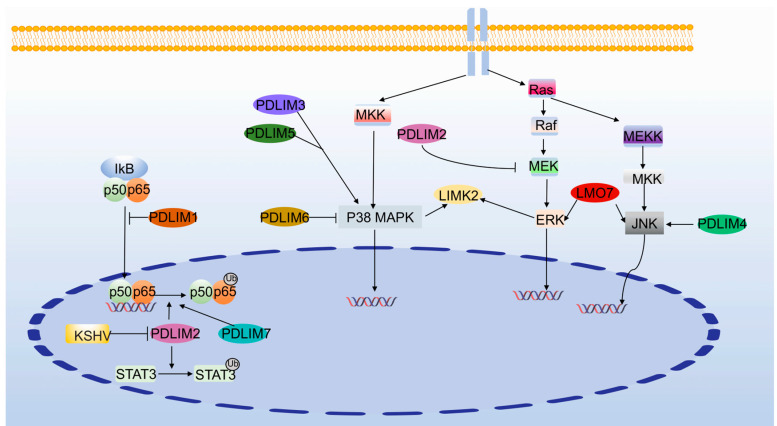
PDLIMs and MAPK and NF-κB signaling pathway.

**Table 1 cancers-15-05042-t001:** ALP subfamily in tumor development and progression.

PDLIMs	Tumor Type	Expression	Effect	Reference
PDLIM1	Glioma	Exogenous suppression	Inhibits tumor invasion	Ahn et al., 2016 [98]
	Breast cancer	Increased	Promotes tumor migration and invasion	Liu et al., 2015 [99]; Gupta et al., 2016 [100]
	Pancreatic cancer	Increased	As a tumor antigen that induces antibody response	Hong 2005 [105]
	Hepatocellular carcinoma	Reduced	Promotes tumor migration	Huang et al., 2020 [101]
	Gastric cancer	Reduced	Promotes tumor progression and cisplatin sensitivity	Tan et al., 2022 [102]
	Colorectal Cancer	Reduced	Inhibits tumor metastasis formation and EMT	Chen et al., 2016 [103]
	Choriocarcinoma	Exogenous suppression	Inhibits tumor cell actin stress fiber formation and focal adhesion assembly	Tamura et al., 2007 [104]
PDLIM2	Breast cancer	Exogenous suppression	Promotes tumor progression and lymphatic metastasis	Ding 2018 [107]; Qu et al., 2010 [108]
	Breast cancer	Increased	Promotes cell migration, cytoskeletal polarization, and EMT	Deevi, Cox, and O’Connor 2014 [109]; Bowe et al., 2014 [110]
	Lung cancer	Reduced	Promotes tumor progression and therapeutic resistance	Sun et al., 2019 [87]; Shi et al., 2020 [111]
	Esophageal squamous cell carcinoma	High PDLIM2 expression group has longer overall survival		Song et al., 2019 [112]
	Hepatocellular carcinoma	Exogenous overexpression	Inhibits tumor malignant phenotype	Jiang et al., 2021 [113]
	Laryngeal squamous cell carcinoma	Reduced	Promotes tumor cell proliferation	Wang et al., 2022 [114]
	Ovarian cancer	Reduced	Promotes tumor pathogenesis	Zhao et al., 2016 [115]
	Metastatic colorectal cancer	Reduced	Promotes tumor metastasis	Oh et al., 2017 [116]
	Castration-Resistant Prostate Cancer	Increased	Promotes tumor growth and invasion	Kang et al., 2016 [91]
	Gastric cancer	Exogenous activation	Promotes tumorigenicity and metastasis	Guo et al., 2016 [117]
PDLIM3	Medulloblastoma	Increased	Unknown	Shou et al., 2015 [118]
	Invasive bladder urothelium carcinoma	Increased	Associated with the unfavorable survival	Feng et al., 2020 [119]
	Bladder cancer	Reduced	Unknown	Lu et al., 2010 [120]
	Thyroid Papillary carcinoma	Reduced	Unknown	Stein et al., 2010 [121]
PDLIM4	Ovarian cancer	Reduced	Associated with aggressive tumor features and poor prognosis	Jia et al., 2019 [122]
	Prostatic carcinoma	Reduced	Promotes carcinogenesis	Vanaja et al., 2009 [123]; Vanaja et al., 2006 [124]; Kolluru et al., 2019 [125]; Vasiljević et al., 2011 [126]
	Thyroid carcinoma	Reduced	Unknown	Patai et al., 2017 [128]
	Kidney cancer	Reduced	Unknown	Morris et al., 2010 [127]
	Breast cancer	Reduced	Promotes tumor progression	Feng et al., 2010 [129] Xu et al., 2012 [130]
	Breast cancer	Exogenous overexpression	Promotes tumor metastasis	Kravchenko et al., 2020 [131]

**Table 2 cancers-15-05042-t002:** Enigma subfamily in tumor development and progression.

PDLIMs	Tumor Type	Expression	Effect	Reference
PDLIM5	Prostatic carcinoma	Increased	Promotes tumorigenesis and migration	Liu et al., 2017 [132]; Shui et al., 2014 [133]
	Thyroid Papillary carcinoma	Increased	Promotes tumor migration, invasion and proliferation	Wei et al., 2018 [135]
	Lung cancer	Increased	Promotes tumor migration and invasion	Shi et al., 2020 [68] Edlund et al., 2012 [137] Zhang et al., 2022 [136] Wu et al., 2023 [138]
PDLIM6	Unknown			
PDLIM7	Acute myeloid leukemia	Increased	An independent risk factor for EFS and OS	Cui et al., 2019 [142]
	Breast cancer	Increased	Correlates with a poor outcome	Kales et al., 2014 [143] Tabariès et al., 2019 [144]
	Colorectal cancer	Exogenous overexpression	Promotes tumor cell survival	Jung et al., 2010 [145]
	Hepatocellular carcinoma	Exogenous overexpression	Promotes tumor cell survival	Jung et al., 2010 [145]
	Thyroid carcinoma	Increased	Promotes carcinogenesis	Firek et al., 2017 [146] Borrello et al., 2002 [147] Kim et al., 2018 [148]
	Gastric cancer	Degradation Inhibited	Contributes to 5-Fu resistance in GC cells	Lu et al., 2023 [149]
	Osteosarcoma	Reduced	Promotes tumor malignant phenotypes	Liu et al., 2014 [150]

**Table 3 cancers-15-05042-t003:** LIM Kinase and LMO7 in tumor development and progression.

PDLIMs	Tumor Type	Expression	Effect	Reference
LIMK1	Gastric cancer	Increased	Promotes tumor growth and metastasis	You et al., 2015 [165]; Li et al., 2010 [166]; Zhang et al., 2016 [167]; H.-Y. Zhang et al., 2017 [168]; J. Zhang et al., 2012 [169]; Su et al., 2016 [170]; Guo et al., 2015 [171]; Li and Chen 2018 [172]; Zeng et al., 2020 [173]; Kang et al., 2021 [174]
	Colorectal cancer	Increased	Promoted tumor development	Liu et al., 2022 [175]; Su et al., 2017 [176]; Liao et al., 2017 [177]; Sun, Li, and Lin 2022 [178]; Zhou et al., 2013 [179]; Sheng et al., 2017 [180]; Chen et al., 2017 [181]
	Colorectal cancer	Exogenous suppression	Promoted tumor development	Zhu et al., 2019 [182]; Hu et al., 2019 [183]
	Breast cancer	Increased	Promoted tumor development	Croft et al., 2005 [184]; McConnell, Koto, and Gutierrez-Hartmann 2011 [185]; Bagheri-Yarmand et al., 2006 [186]; Lagoutte et al., 2016 [187]; Yan et al., 2021 [188]; Shi et al., 2021 [189]; Li et al., 2014 [190]; Shahi et al., [191]; Fu et al., 2018 [192]; Zhao, Li, and Fang 2019 [193]; Li, Hu, et al., 2017 [194]; Li, Wang, et al., 2017 [195]; Li et al., 2018, 1 [196] J.
	Prostatic carcinoma	Exogenous overexpression	Promoted tumor development	Davila et al., 2007 [197]; Moretti et al., 2011 [198]; Cai et al., 2015 [199]; Ngalame et al., 2016 [200]; Bhardwaj et al., 2014, 1 [201]; Mardilovich et al., 2015 [202]
	Lung cancer	Increased	Promoted tumor development	Lu et al., 2021 [203]; Cai et al., 2015 [204]; Guo et al., 2016 [205]; Zhang et al., 2021 [206]; Kang et al., 2017 [207]; Wan et al., 2014 [208]; Chen et al., 2013 [209]
	Osteosarcoma	Increased	Promoted tumor development	Yang et al., 2018 [210]; H.-S. Zhang et al., 2014 [211]; Yoshizawa et al., 2019, 1 [212]; H. Zhang et al., 2011 [213]
	Uterine cervix carcinoma	Increased	Promoted tumor development	Chhavi et al., 2010 [214]; Yang et al., 2021 [215]; Xu et al., 2021 [216]
	Hepatocellular carcinoma	Exogenous overexpression	Promoted tumor development	Ding et al., 2023 [217]; Wang et al., 2020 [218]; Pan et al., 2021 [219]
	Melanoma	Exogenous suppression	Promotes tumor invasion	Lee et al., 2012 [220]
	Oral cancer	Exogenous suppression	Inhibits tumor proliferation, migration and EMT	Shi et al., 2019 [221]
	Thyroid Papillary carcinoma	Exogenous suppression	Inhibits tumor progression	Xiong, Zhang, and Kebebew 2014 [223]
	Cutaneous squamous cell carcinoma	Exogenous suppression	Inhibits tumor progression	Zhou et al., 2014 [224]
	Glioblastoma	Exogenous suppression	Inhibits tumor progression	Peng et al., 2020 [222]
	Pancreatic adenocarcinoma	Exogenous activation	Promotes tumor growth and metastasis	Wang et al., 2021 [225]
	Ovarian cancer	Increased	Promotes tumor migration	Zhang, Gan, and Zhou 2012 [226]; Chen et al., 2014 [227]
LIMK2	Colorectal cancer	Exogenous overexpression	Inhibits tumor cell proliferation	Zhang et al., 2019 [231]
	Colorectal cancer	Increased	Promotes tumor development	Aggelou et al., 2018 [230]
	Colorectal cancer	Reduced	Promotes tumor development	Yue Zhang et al., 2018 [228]; Lourenço et al., 2014 [229]
	Castration resistant prostate cancer	Increased	Promotes tumor initiation, progression, and poor prognosis	Nikhil et al., 2019 [232]
	Prostate cancer	LIMK regulates other molecules	Promotes tumor progression and drug resistance	Nikhil, Kamra, et al., 2021 [246]; Nikhil et al., 2021 [247]; Sooreshjani et al., 2021 [248]
	Triple negative breast cancer	LIMK regulates other molecules	Promotes tumor migration	Malvi et al., 2020 [237]
	Breast cancer	Exogenous suppression	Inhibits tumor migration	Shahi et al., 2017 [191]; Malvi et al., 2020 [237]
	Non-small-cell lung cancer	Exogenous activation	Inhibits tumor proliferation	Xu et al., 2019 [238]
	Non-small-cell lung cancer	Exogenous overexpression	Promotes tumor proliferation and drug resistance	Niu et al., 2017 [233]; Su et al., 2022 [234]
	Osteosarcoma	Exogenous activation	Promotes tumor migration	Ren et al., 2019 [235]; Wang et al., 2017 [239]
	Neuroblastoma	Increased	Promotes tumor drug resistance	Po’uha et al., 2010 [240]; Gamell et al., 2013 [241]
	Pancreatic adenocarcinoma	Exogenous suppression	Inhibits tumor development	Rak et al., 2014 [243]
	Bladder cancer	Increased	Promotes tumor proliferation, migration, and invasion	Wang et al., 2019 [244]
	Thyroid carcinoma	Reduced	Promotes tumor migration	Hsu et al., 2010 [245]
	Esophageal cancer	Reduced	Promotes tumor migration	Hsu et al., 2010 [245]
LMO7	Breast cancer	Increased	Promotes tumor migration	Hu et al., 2011 [151]; Tanaka-Okamoto et al., 2009, 7 [152]; Nakamura et al., 2011 [153]
	Lung cancer	Exogenous suppression	Promoted tumor development	Wu et al., 2017 [154]; Yang et al., 2022 [155]; Li et al., 2021 [156]
	Cervical carcinoma	Exogenous overexpression	Inhibits tumor cell proliferation	Tzeng et al., 2018 [157]
	Colorectal cancer	Exogenous overexpression	Inhibits tumor cell proliferation	Tzeng et al., 2018 [157]
	Pancreatic cancer	Increased	Promotes tumor progression and metastasis	Lin et al., 2021 [158]
	Hepatocellular carcinoma	Exogenous overexpression	Promotes tumor invasion	Nakamura et al., 2005 [72]

**Table 4 cancers-15-05042-t004:** Abbreviations in this article.

ATC	Undifferentiated thyroid cancer
BC	Breast cancer
BCA	Bladder carcinoma
BRCA	Breast invasive carcinoma
CC	Choriocarcinoma
CCA	Cervical carcinoma
CRC	Colorectal cancer
EMT	Epithelial–mesenchymal transition
EC	Esophageal cancer
FVPTC	Follicular papillary thyroid carcinoma
GC	Gastric cancer
GCC	Hepatocarcinoma
LSCC	Laryngeal squamous cell carcinoma
LUAD	Lung adenocarcinoma
LC	Lung cancer
NSCLC	Non-small-cell lung cancer
OS	Osteosarcoma
OV	Ovarian Cancer
PAAD	Pancreatic adenocarcinoma
PTC	Papillary thyroid carcinoma
PDTC	Poorly differentiated thyroid carcinoma
PC	Prostatic carcinoma
THCA	Thyroid carcinoma

## Data Availability

The data can be shared up on request.

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
