# Peer review of "PDZ and LIM Domain-Encoding Genes: Their Role in Cancer Development"

_cancers, 2023, doi:10.3390/cancers15205042_

Round 1
Reviewer 1 Report
In this Review, the Authors presented evidence concerning PDZ-LIM family proteins (PDLIMs), their structural domains (PDZ and LIM), and their role in various biological processes, including cell signaling and tumor development. The main focus is quite clear. The text highlights the importance of these domains in various physiological processes, including cell differentiation, proliferation, migration, and maintenance of cellular structural integrity. The mention of their involvement in tumor formation and development emphasizes their significance in biology.
The Review can be published. There are however several concerns:
Supplement.
Please, move list of abbreviations into the main text.
Subsection
2.1. PDZ domain
The text discusses the evolutionary aspects of the PDZ domain, but the relevance and implications of this information could be clarified. How does the PDZ domain's robustness to mutation relate to its biological significance or applications? Providing more context would enhance the chapter.
Subsection
3. PDLIMs and signaling pathways
Although the text presents a wealth of information about the interactions between PDLIMs and various signaling pathways, it lacks contextual explanations of why these interactions are significant or how they relate to broader biological processes. Providing context and explanations for the relevance of these interactions would enhance the reader's understanding
While the depth of information can be beneficial for experts, it may overwhelm readers who do not have an in-depth background in molecular biology and cell signaling. Balancing the level of detail for different audiences would improve accessibility.
Please, provide an explanatory figure to this chapter. In the present form it is difficult for perception.
Conclusion:
The text discusses multiple aspects of PDZ-LIM proteins and their role in cancer, which can be overwhelming for readers who are not experts in the field. Simplifying the language and providing more concise explanations would enhance readability.
The text introduces several questions and areas of further study, such as the triggers for PDZ-LIM protein dysregulation in tumors and potential synergistic effects among protein subfamilies. However, these questions are scattered throughout the text and could benefit from a more organized structure
And… the final concern
Please, improve the text thought the Review by making it less technical. Current version is densely packed with information thus making it challenging for readers to navigate and locate specific details. Improved organization and small conclusions in the end of each chapter would enhance readability. The text discusses multiple aspects of PDZ-LIM proteins and their role in cancer, which can be overwhelming for readers who are not experts in the field. Simplifying the language and providing more concise explanations would attract more citations.
English is qute good. At the same time Syle needs polishing.
Author Response
Dear Editors and Reviewers:
Thank you for your letter and for the reviewers’ comments concerning our manuscript entitled “PDZ and LIM domain-encoding genes: their role in cancer development (ID: cancers-2631084)”. Those comments are all valuable and very helpful for revising and improving our manuscript, as well as the important guiding significance to our researches. We have considered comments carefully and have made correction which we hope meet with approval. According to editors’ suggestion, we confirm that all of figures have copyright and modify citations in the correct format. Revised portions are marked in yellow in the manuscript. The main corrections in the paper and the responds to the reviewers’ comments are as follows:
Response to Review 1#
In this Review, the Authors presented evidence concerning PDZ-LIM family proteins (PDLIMs), their structural domains (PDZ and LIM), and their role in various biological processes, including cell signaling and tumor development. The main focus is quite clear. The text highlights the importance of these domains in various physiological processes, including cell differentiation, proliferation, migration, and maintenance of cellular structural integrity. The mention of their involvement in tumor formation and development emphasizes their significance in biology.
Thank you so much for your recognition and professional comments on our article. As you are concerned, there are several issues that need to be addressed. Based on your suggestions, we have made extensive revisions on the previous manuscript, changes in the manuscript are highlighted in yellow. In the point-to-point letter. The reviewer's comments are listed in italics, specific questions are numbered, and our responses are provided in normal font as follows:
-
Please, move list of abbreviations into the main text.
Reply: Thank you for your suggestion. We have placed the abbreviations and Table 4 in the main text, see page 30.
(2) 2.1. PDZ domain
The text discusses the evolutionary aspects of the PDZ domain, but the relevance and implications of this information could be clarified. How does the PDZ domain's robustness to mutation relate to its biological significance or applications? Providing more context would enhance the chapter.
Reply: We sincerely appreciate your valuable suggestion. After reviewing a large amount of literature, relevant researchers in this field have concluded that the robustness of PDZ evolution is an ideal mode to support the high-speed evolution of interaction networks. We have supplemented the relevant explanations in the article, see line page 5-8. The PDZ domain binds to the substrate through seven amino acids at its end, and researches suggest that mutations in this region can alter the structure of the PDZ domain and the polarity of amino acids in the substrate binding region, leading to enhanced or disappeared binding to the substrate. The specific research content has been elaborated in detail in reference 16 (PMID: 18828675), and we have made some modifications in this section to facilitate readers for better understanding. So in the revised manuscript we will no longer expand the content on these topics.
(3)3. PDLIMs and signaling pathways
Although the text presents a wealth of information about the interactions between PDLIMs and various signaling pathways, it lacks contextual explanations of why these interactions are significant or how they relate to broader biological processes. Providing context and explanations for the relevance of these interactions would enhance the reader's understanding
While the depth of information can be beneficial for experts, it may overwhelm readers who do not have an in-depth background in molecular biology and cell signaling. Balancing the level of detail for different audiences would improve accessibility.
Please, provide an explanatory figure to this chapter. In the present form it is difficult for perception.
Reply: Thank you very much for your suggestion. Based on your suggestion, we have re-written this section. We have integrated a large number of literature and summarized the patterns in which different PDLIMs interact with signaling pathways and their impact on biological processes. And based on the summarized information, the signal pathways involved in PDLIMs are plotted and presented in Figure 4/5/6 in the main text of the article, as shown on Page 8-11.
(4)Conclusion:
The text discusses multiple aspects of PDZ-LIM proteins and their role in cancer, which can be overwhelming for readers who are not experts in the field. Simplifying the language and providing more concise explanations would enhance readability.
The text introduces several questions and areas of further study, such as the triggers for PDZ-LIM protein dysregulation in tumors and potential synergistic effects among protein subfamilies. However, these questions are scattered throughout the text and could benefit from a more organized structure.
Reply: This is a good suggestion, and we have made corresponding modifications to the manuscript, simplifying and merging the language, and highlighting the key points. In the summary section, the logic of the article is highly summarized, including the physiological importance of PDLIMs, the relationship between their disorders and tumors, the therapeutic significance of targeting PDLIMs, and future research directions. Please refer to Page 30-31.
(5)And… the final concern
Please, improve the text thought the Review by making it less technical. Current version is densely packed with information thus making it challenging for readers to navigate and locate specific details. Improved organization and small conclusions in the end of each chapter would enhance readability. The text discusses multiple aspects of PDZ-LIM proteins and their role in cancer, which can be overwhelming for readers who are not experts in the field. Simplifying the language and providing more concise explanations would attract more citations.
Reply: We do our best to revise the manuscript, merge duplicate content, eliminate irrelevant information, condense sentences, and make the article as concise and readable as possible. add conclusions in appropriate positions to make the article more readable. These changes in text adjustments will not affect the content and framework of the article. In addition, we have plotted and illustrated the signaling pathways that PDLIMs involved, making it easier for readers to understand. At the same time, we have removed the chapter of “PDLIMs and other diseases” in the revised version, and simplified and summarized the content in the signal pathway section. We hope that the revised version of the manuscript is more accessible for reviewers and readers. The revised parts of the article have been marked in yellow. We sincerely thank the reviewers for their enthusiastic work and hope that the corrected manuscript can improve the quality of the article.
Reviewer 2 Report
The provided article discusses the role of PDLIMs (PDZ and LIM domain-containing proteins) as scaffold proteins in regulating signaling cascades and their involvement in various physiological processes, particularly in the context of tumor development. While the article contains valuable information, it requires major revisions to enhance clarity, organization, and scientific rigor. Here are suggested improvements for a major revision:
- Please do not use abbreviations in the title
- Introduction:
- The introduction should provide context for the role of scaffold proteins in cellular signaling and their importance.
- Include a clear research question or objective for the review.
- Section 2: Structural Features of the PDLIMs:
- Break this section into subsections for PDZ domain, LIM domain, and other domains for better organization.
- Provide more context and examples for each domain's function.
- Clarify the significance of these domains in cellular processes.
- Section 3: PDLIMs and Signaling Pathways:
- Organize this section into subsections based on the signaling pathways discussed (integrin-related, TGF-β, MAPK, etc.).
- Include figures or diagrams to illustrate the interactions and pathways involving PDLIMs.
- Clarify how PDLIMs modulate these pathways and their functional consequences.
- Section 4: PDLIMs and Tumor:
- Divide this section into subsections for each PDLIM family member (PDLIM1, PDLIM2, PDLIM3, PDLIM4, etc.) and their roles in different types of tumors.
- Provide more context and mechanistic details for each PDLIM's role in tumor development.
- Include tables or figures summarizing the findings for each PDLIM family member.
- Discussion
- Please include and discuss the following: PMID: 37446024; PMID: 37685983
Certainly, in addition to the content and structural changes suggested, it's important to shorten the overall length of the article to make it more concise and accessible. Here are some strategies to reduce the overall length:
- Paragraph Summaries: Review each paragraph to ensure it contains only essential information. Eliminate repetitions or unnecessary details.
- Eliminate Redundant Information: Avoid repeating concepts or data already presented earlier. Each piece of information should add value.
- Sentence Condensation: Condense long and complex sentences into shorter, clearer ones.
- Remove Unnecessary Examples: If you've provided examples to clarify a point, try to keep only the most relevant and effective ones.
- Focus on Key Aspects: Concentrate on the most crucial and relevant aspects of your topic. Remove peripheral details.
- Reduce Citations: If possible, reduce the number of citations, keeping only those essential to support your main claims.
- Limit Subsections: Very detailed sections and subsections may be combined or simplified.
Shortening the length will make the article more accessible and increase the effectiveness of communicating key information to readers.
By addressing these points, the article can be significantly improved in terms of organization, clarity, and scientific rigor, making it more informative and engaging for readers.
Author Response
Dear Editors and Reviewers:
Thank you for your letter and for the reviewers’ comments concerning our manuscript entitled “PDZ and LIM domain-encoding genes: their role in cancer development (ID: cancers-2631084)”. Those comments are all valuable and very helpful for revising and improving our manuscript, as well as the important guiding significance to our researches. We have considered comments carefully and have made correction which we hope meet with approval. According to editors’ suggestion, we confirm that all of figures have copyright and modify citations in the correct format. Revised portions are marked in yellow in the manuscript. The main corrections in the paper and the responds to the reviewers’ comments are as follows:
Response to Review 2#
The provided article discusses the role of PDLIMs (PDZ and LIM domain-containing proteins) as scaffold proteins in regulating signaling cascades and their involvement in various physiological processes, particularly in the context of tumor development. While the article contains valuable information, it requires major revisions to enhance clarity, organization, and scientific rigor. Here are suggested improvements for a major revision:
Thank you so much for your recognition and professional comments on our article. As you are concerned, there are several issues that need to be addressed. Based on your suggestions, we have made extensive revisions on the previous manuscript, changes in the manuscript are highlighted in yellow. In the point-to-point letter. The reviewer's comments are listed in italics, specific questions are numbered, and our responses are provided in normal font as follows:
(1)Please do not use abbreviations in the title.
Reply: Thank you for your suggestion. PDZ and LIM are the name of protein domain, they
are not abbreviations.
(2) Introduction:
- The introduction should provide context for the role of scaffold proteins in cellular signaling and their importance.
- Include a clear research question or objective for the review.
Reply: Thank you very much for your valuable suggestions. We have made modifications to the introduction section, explaining the role and importance of scaffold proteins in signal transduction, and explaining the research purpose of this review. Please refer to page 3-4.
(3) Section 2: Structural Features of the PDLIMs:
- Break this section into subsections for PDZ domain, LIM domain, and other domains for better organization.
- Provide more context and examples for each domain's function.
- Clarify the significance of these domains in cellular processes.
Reply: This is a good suggestion. Our article has been divided into subsections for PDZ domain, LIM domain, and other domains. And large numbers of literatures have been consulted to expand on examples of different domains and elucidate their impact on cell function. See Page 5-8.
(4)Section 3: PDLIMs and Signaling Pathways:
- Organize this section into subsections based on the signaling pathways discussed (integrin-related, TGF-β, MAPK, etc.).
- Include figures or diagrams to illustrate the interactions and pathways involving PDLIMs.
- Clarify how PDLIMs modulate these pathways and their functional consequences.
Reply: Thank you for your valuable suggestion. Based on your suggestion, we have made modifications to this section to make it more organized. We have divided the content into different sections according to different signal pathways, clarifying the ways in which PDLIMs affect signal pathways and their functional impacts. The content of this section is placed in the main text of the article as shown in Figure 4/5/6, as shown on Page 8-11.
(5) Section 4: PDLIMs and Tumor:
- Divide this section into subsections for each PDLIM family member (PDLIM1, PDLIM2, PDLIM3, PDLIM4, etc.) and their roles in different types of tumors.
- Provide more context and mechanistic details for each PDLIM's role in tumor development.
- Include tables or figures summarizing the findings for each PDLIM family member.
Reply: Your suggestion is very valuable for improving the quality of this section. We have previously divided this section into several sections based on different PDLIMs and plotted them into tables, as shown in Tables 1, 2, and 3. We have expanded the content of the article to provide more examples and background information on the role of different PDLIMs in different tumors. Please refer to Page 11-27.
(6)Discussion
- Please include and discuss the following: PMID: 37446024; PMID: 37685983. Certainly, in addition to the content and structural changes suggested, it's important to shorten the overall length of the article to make it more concise and accessible. Here are some strategies to reduce the overall length:
- Paragraph Summaries: Review each paragraph to ensure it contains only essential information. Eliminate repetitions or unnecessary details.
- Eliminate Redundant Information: Avoid repeating concepts or data already presented earlier. Each piece of information should add value.
- Sentence Condensation: Condense long and complex sentences into shorter, clearer ones.
- Remove Unnecessary Examples: If you've provided examples to clarify a point, try to keep only the most relevant and effective ones.
- Focus on Key Aspects: Concentrate on the most crucial and relevant aspects of your topic. Remove peripheral details.
- Reduce Citations: If possible, reduce the number of citations, keeping only those essential to support your main claims.
- Limit Subsections: Very detailed sections and subsections may be combined or simplified.
Reply: We have cited these two articles (PMID: 37446024; PMID: 37685983). We do our best to revise the manuscript, merge duplicate content, eliminate irrelevant information, condense sentences, and make the article as concise and readable as possible. add conclusions in appropriate positions to make the article more readable. These changes in text adjustments will not affect the content and framework of the article. The revised text of the manuscript has been marked in yellow. We sincerely thank you for your enthusiastic work and hope that the manuscript will be approved.
Round 2
Reviewer 2 Report
The authors have conducted a sufficiently good revision to deem the article worthy of publication.